# Hazard Function Analysis of Recurrence in Patients with Curatively Resected Lung Cancer: Results from the Japanese Lung Cancer Registry in 2010

**DOI:** 10.3390/cancers14205119

**Published:** 2022-10-19

**Authors:** Yoshikane Yamauchi, Masafumi Kawamura, Jiro Okami, Yasushi Shintani, Hiroyuki Ito, Takashi Ohtsuka, Shinichi Toyooka, Takeshi Mori, Shun-ichi Watanabe, Hisao Asamura, Masayuki Chida, Shunsuke Endo, Mitsutaka Kadokura, Ryoichi Nakanishi, Etsuo Miyaoka, Hidemi Suzuki, Ichiro Yoshino, Hiroshi Date

**Affiliations:** 1Department of Surgery, Teikyo University School of Medicine, Tokyo 173-8605, Japan; 2Department of General Thoracic Surgery, Osaka International Cancer Institute, Osaka 541-8567, Japan; 3Department of General Thoracic Surgery, Osaka University Graduate School of Medicine, Osaka 565-0871, Japan; 4Department of Thoracic Surgery, Kanagawa Cancer Center, Yokohama 241-8515, Japan; 5Division of Thoracic Surgery, Department of Surgery, Jikei University School of Medicine, Tokyo 105-8461, Japan; 6Department of Thoracic Surgery, Okayama University Hospital, Okayama 700-8558, Japan; 7Department of Thoracic Surgery, Japanese Red Cross Kumamoto Hospital, Kumamoto 861-8520, Japan; 8Department of Thoracic Surgery, National Cancer Center Hospital, Tokyo 104-0045, Japan; 9Division of General Thoracic Surgery, Department of Surgery, School of Medicine, Keio University, Tokyo 160-8582, Japan; 10Department of General Thoracic Surgery, Dokkyo Medical University, Shimotsuga-gun, Tochigi 321-0293, Japan; 11Department of Thoracic Surgery, Jichi Medical University, Shimotsuke 329-0498, Japan; 12Division of Chest Surgery, Department of Surgery, Showa University School of Medicine, Tokyo 142-8555, Japan; 13Department of Oncology, Immunology and Surgery, Nagoya City University Graduate School of Medical Sciences, Nagoya 467-8602, Japan; 14Department of Mathematics, Tokyo University of Science, Tokyo 162-8601, Japan; 15Department of General Thoracic Surgery, Graduate School of Medicine, Chiba University, Chiba 260-8670, Japan; 16Department of Thoracic Surgery, Kyoto University Graduate School of Medicine, Kyoto 606-8501, Japan

**Keywords:** lung cancer recurrence, hazard function, postoperative surveillance

## Abstract

**Simple Summary:**

To optimize postoperative surveillance of lung cancer patients, we investigated the hazard function of tumor recurrence in patients with completely resected lung cancer. Using the records of the 2010 Japanese Joint Committee of Lung Cancer Registry, the risk of postoperative recurrence was analyzed using a cause-specific hazard function in patients who underwent lobectomy to completely resect pathological stage I–III lung cancer. The hazard function for recurrence exhibited a peak at approximately 9 months after surgery, followed by a tapered plateau-like tail extending to 60 months. The peak risk for intrathoracic recurrence was approximately two-fold higher compared with that of extrathoracic recurrence. When considered together with the results of the subgroup analysis, the characteristics of the postoperative tumor recurrence hazard in a large cohort of lung cancer patients may be useful for improving stage-related management of postoperative surveillance.

**Abstract:**

To optimize postoperative surveillance of lung cancer patients, we investigated the hazard function of tumor recurrence in patients with completely resected lung cancer. We analyzed the records of 12,897 patients in the 2010 Japanese Joint Committee of Lung Cancer Registry who underwent lobectomy to completely resect pathological stage I–III lung cancer. The risk of postoperative recurrence was determined using a cause-specific hazard function. The hazard function for recurrence exhibited a peak at approximately 9 months after surgery, followed by a tapered plateau-like tail extending to 60 months. The peak risk for intrathoracic recurrence was approximately two-fold higher compared with that of extrathoracic recurrence. Subgroup analysis showed that patients with stage IIIA adenocarcinoma had a continuously higher risk of recurrence compared with patients with earlier-stage disease. However, the risk of recurrence in patients with squamous cell carcinoma was not significantly different compared with that more than 24 months after surgery, regardless of pathological stage. In conclusion, the characteristics of postoperative tumor recurrence hazard in a large cohort of lung cancer patients may be useful for determining the time after surgery at which patients are at the highest risk of tumor recurrence. This information may improve stage-related management of postoperative surveillance.

## 1. Introduction

Lung cancer is one of the most commonly diagnosed malignancies and the leading cause of cancer death worldwide [1]. Unfortunately, only 13.5% of new cases are diagnosed as surgically resectable pathological stage I–IIIA non-small cell lung cancer (NSCLC) [2]. However, lung cancer screening will likely increase the incidence of early-stage lung cancer detected by low-dose computed tomography (CT), because the effective mass screening of high-risk groups by CT must be beneficial according to a randomized trial [3]. Therefore, an increase in resectable lung cancer cases is anticipated. The 5-year survival rate after complete resection of NSCLC is approximately 60%, mainly because of postoperative recurrence [4]. Therefore, the organization of structured follow-up protocols is required to improve overall and disease-free survival through early detection and treatment of tumor recurrence [5]. Moreover, if a strict follow-up protocol is implemented, general medical care will improve for patients who require rigorous follow-up. Furthermore, symptomatic patients whose recurrence is diagnosed during follow-up achieve a significant survival advantage. These findings support the implementation of intensive follow-up after complete resection [6]. However, medical resources are limited, and it is not practical to conduct rigorous follow-ups on all patients who have undergone surgery. Therefore, an appropriate postoperative surveillance protocol should be developed to effectively utilize limited resources. To establish an optimal protocol, more information is required on who is more likely to experience recurrence, when, and at what site after complete resection of lung cancer. This can be solved using a hazard function that describes the risk of recurrence among the remaining patients at each time.

We previously found that the hazard function may be useful for selecting patients at high risk of recurrence [7]. However, this study was a retrospective analysis of data from a single institution, which limited the content of the analysis. The study enrolled approximately 1400 patients for 7 years. However, when sorted by the site of recurrence, the number of eligible cases became very small and consequently was insufficient to identify an individualized follow-up protocol. Nonetheless, we considered that lengthening the study period to enroll more patients would have introduced several confounding factors because the treatment of lung cancer is changing very rapidly [8,9,10] and its impact cannot be eliminated when the study period is extended too much. To overcome these limitations, we conducted the present study through an analysis of a national registry database comprising records for the year 2010.

Here, we evaluated the hazard function of tumor recurrence in patients with completely resected lung cancer and generated a smoothing curve from the risk of recurrence to characterize the recurrence pattern according to clinicopathological factors.

## 2. Materials and Methods

### 2.1. Registry

The Japanese Joint Committee of Lung Cancer Registry used a nationwide registry to retrospectively study patients who underwent surgery for lung cancer. The committee invited 629 teaching hospitals certified by the Japanese Board of General Thoracic Surgery to participate. The review board of Osaka University Medical Hospital approved this study (approval No. 15321), which was conducted in accordance with ethical guidelines for epidemiological studies [11,12]. The study was registered with the University Hospital Medical Information Network-Clinical Trials Registry (identification No. 000020215). The demographics of the registry have already been published [13]. The following items of information were extracted from the registry for use in this study: age, sex, performance status, smoking history, preoperative lung function (%VC and FEV1/FVC), type of surgery, surgery time, pathological T factor, tumor size, pathological N factor, pathological stage, histology, pleural infiltration, adjuvant therapy, and prognosis, including the date and the site of tumor recurrence. The initial data were accumulated during 2010, and pathological stages were therefore described according to the TNM Classification, 7th edition [14].

### 2.2. Patients

Inclusion criteria were as follows: (1) pathological diagnosis of any type of primary lung cancer; (2) surgery with curative intent performed between 1 January 2010 and 31 December 2010; and (3) complete surgical resection (R0).

The study flowchart is shown in Figure 1a. Patients who received neoadjuvant treatment before surgery were excluded. Among 18,973 patients in the nationwide registry study, 12,897 were included. The collection of patients’ prognostic information was performed online from 1 January 2016 to 31 October 2016.

### 2.3. Definition of Recurrence

The surveillance program was not consistent throughout this study because this was a retrospective study in which many medical institutions participated. Generally speaking, recurrence was diagnosed according to the results of physical and imaging studies and was pathologically confirmed by invasive procedures if necessary. The recurrence date was determined as the date on which recurrence was confirmed by the clinical and radiological findings. We assumed that each hospital had its own institutional oncology review board that determined the onset and site of recurrence when recurrence was detected in the enrolled patients. Recurrence data were subsequently designated in the database. If two recurrence sites were found at the same time, both sites were registered as recurrence sites. However, if two recurrence sites were detected at different times, the site identified earlier was registered as the recurrence site. Although numerous sites harbored recurrences, these were categorized as intrathoracic recurrence (IR) and extrathoracic recurrence (ER). IR included recurrence at the surgical stump, at mediastinal or hilar lymph nodes, pleural dissemination, and in the residual lung. ER included recurrence in bone, brain, adrenal gland, liver, and extrathoracic lymph nodes.

### 2.4. Statistical Analysis

The hazard function was defined as the number of subjects who had not yet failed divided by the number of subjects at risk [15]. The goal of this function in this study was to model a participant’s chance of recurrence. The hazard function depends on the number of failures in a given small time period [∆t] and the number of cases subjected to failure.
$$h\;\left(t\right)=\frac{\mathrm{Number}\;\mathrm{of}\;\mathrm{recurrent}\;\mathrm{cases}\;\mathrm{in}\;a\;\mathrm{month}}{\mathrm{Number}\;\mathrm{of}\;\mathrm{nonrecurrent}\;\mathrm{cases}\;\mathrm{just}\;\mathrm{before}\;\mathrm{the}\;\mathrm{time}}$$

In this study, the hazard function was defined as the above formula. The hazard function describes the instantaneous failure rate for the event to occur if the individual survives until a certain time. The failure rate is herein described as the hazard rate. The time scale was discretized by 1 month, and all hazard rates were measured as “events/patients at risk per month interval” (Figure 1b). The monthly hazard rates were estimated using the Epanechnikov kernel smoothing method [16] because the hazard rate estimates were unstable due to a random fluctuation, and a smoothed curve is better for understanding the hazard rate patterns. SPSS 25 software (IBM Corp., Armonk, NY, USA) was used to generate the hazard curves.

## 3. Results

The clinicopathological characteristics of 12,897 patients are shown in Table 1. The most frequent procedure was lobectomy (*n* = 12,335, [95.6%]), followed by bilobectomy (*n* = 335, [2.6%]) and pneumonectomy (*n* = 227, [1.8%]). All patients were designated as complete resection, and 4454 patients (34.5%) received adjuvant treatment, mainly chemotherapy. Adenocarcinoma (ADC) and squamous cell carcinoma (SqCC) were the most frequent tumors (*n* = 9004 and *n* = 2691 (9.8% and 20.9%), respectively).

Figure 2a shows the monthly recurrence hazard function for the entire population. The hazard function demonstrated that the major peak of recurrence was 9 months after surgery. Subsequently, the curve declined sharply until 24 months and then extended as a tapered plateau-like tail to 60 months. The hazard curve for IR (Figure 2b) was approximately two-fold higher compared with that of ER (Figure 2c) from the beginning of treatment to 60 months after surgery. Furthermore, the IR curve peaked approximately 9 months after surgery, declined to approximately 60% of its peak at 24 months, and continued to slowly decline until 60 months. All IR sites showed similar hazard patterns for recurrence (Appendix A). In contrast, the hazard curve for ER peaked at approximately 9 months, then quickly declined to approximately 20% of its peak at 24 months, and subsequently plateaued until 60 months. The results of the hazard function analyses of bone and brain metastases, in particular, (Figure 2d and Figure 2e, respectively), showed that the curve reached 25% of its peak after approximately 24 months and subsequently plateaued. The findings were similar for other ER sites (Appendix A), and the hazard patterns of IR and ER of patients with ADC or SqCC were similar to those of the entire patient population (Appendix A).

Figure 3a shows a comparison of the recurrence hazard according to tumor histology. The curve peaked at 6–12 months after surgery in all cases, decreased until 24 months, and exhibited a tapered plateau-like tail extending to 60 months after surgery. The values of the heights of the peaks, in descending order, were as follows: small cell carcinoma, sarcoma, adenosquamous cell carcinoma, large cell neuroendocrine carcinoma, SqCC, and ADC. The approximately flat curve of carcinoid tumors lacked a distinct peak.

Figure 3b shows a comparison of recurrence hazards associated with pathological stages. The hazard curve for stage IA was approximately flat with no clear peak, and those for stage IB and stage II peaked at approximately 9 months after surgery. The heights of the peaks in descending order were stage IIB, stage IIA, and stage IB, which subsequently decreased until approximately 24 months after surgery. The three overlapping curves slowly decreased until 60 months after surgery. The shape of the curve for stage IIIA was similar. However, the curve consistently maintained an approximately two-fold higher recurrence hazard compared with those of stages IB and II until 60 months after surgery.

Figure 4a presents a comparison of the recurrence hazard associated with the pathological stages of patients with ADC. The hazard curve for stage IA was approximately flat with no clear peak. The hazard curves for stage IB and stage II peaked approximately 12 months after surgery. The heights of the hazard peaks in descending order were stage IIB, stage IIA, and stage IB, each of which subsequently decreased until approximately 24 months. The curves of stage IIA and stage IIB overlapped and subsequently decreased until 60 months after surgery. The curves of stage IA, stage IB, stage IIA, and stage IIB overlapped at approximately 48 months and subsequently decreased at a slower rate until 60 months after surgery. The shape of the curve for stage IIIA was similar; however, it consistently maintained an approximately two-fold higher recurrence hazard compared with that of stage II until 60 months after surgery.

Figure 4b compares the recurrence hazard of the pathological stages of patients with SqCC. The hazard curve for stage IA was approximately flat with no clear peak. In contrast, the hazard curves for stage IB, stage IIA, stage IIB, and stage IIIA peaked approximately 6 months after surgery. The heights of the hazard peaks in descending order were stage IIIA, stage IIB, stage IIA, and stage IB. The hazards, which subsequently decreased until approximately 24 months after surgery, overlapped at approximately 24 months and subsequently flattened with no clear peak.

Figure 5a compares the recurrence hazard with and without adjuvant chemotherapy. The hazard peak was delayed by approximately 1.7 months in the patients with adjuvant chemotherapy. The hazard peak was next investigated by stratification of all patients according to stage (Figure 5b–e). The peak was delayed by approximately 3 months in the adjuvant chemotherapy group except for stage IIB. The height of the peak of the adjuvant chemotherapy group was higher for those with stage IB, although it was similar to that of stage II and lower than that of stage IIIA.

## 4. Discussion

The primary finding of the present study is that the time trends of the recurrence hazard of patients with completely resected lung cancer displayed sharp transitions during follow-up after surgery. As we previously reported [7], the hazard function provides information about the transition of recurrence probabilities during follow-up. The peaks of recurrence shown here are consistent with those reported by others [7,17,18,19,20,21]. Moreover, the shapes of the curves of recurrence hazard rates exhibited unique patterns, which depended on clinical and pathological factors. To the best of our knowledge, this is the largest analysis of postoperative recurrence patterns of patients with completely resected lung cancer. Furthermore, while lung cancer treatment strategies are rapidly updated, the results of a study such as this are easily influenced by changes in treatment strategies. However, the present study analyzed patients’ records accumulated during 2010 when treatments protocol were consistent. Furthermore, only Japanese medical institutions participated in this registry. In Japan, a uniform national health insurance system has been introduced [22], and due to the restrictions of this health insurance system, treatment should be conducted according to clinical guidelines annually updated by the Japan Lung Cancer Society [23]. Therefore, we believe that the reliability of the generalized results is high because it is difficult to have large differences in treatment details among medical institutions.

The hazard function may be useful for selecting patients at high risk of recurrence and may provide information to optimize individual follow-up examinations. For example, the peak hazard increases as the pathological stage increases, suggesting that patients with higher stages should undergo detailed surveillance testing within a year. Furthermore, only stage IIIA remained higher than other stages, although the hazard height approached it over time, suggesting that surveillance of stage IIIA may need to be frequently performed even more than 2 years after surgery. However, histological types such as sarcoma, small cell lung cancer, adenosquamous cell carcinoma, and large cell carcinoma had higher hazard peaks than ADC and SqCC, suggesting that they should be closely examined within 1 year. Nonetheless, as time goes by, the hazard height approaches the level of other stages, indicating that surveillance need not be changed based on histological type alone. These features of the hazard functions may be useful in planning postoperative surveillance, since even early-stage lung cancer cases, which are easier to detect with low-dose CT screening, can recur postoperatively [24,25].

For patients without recurrence 2 years after surgery, routine screening using a chest CT may be beneficial because of the high frequency of IR. In contrast, the frequency of ER is low in this population, and therefore routine screening for ER may not be useful. In particular, head magnetic resonance imaging, bone scintigraphy, and positron emission tomography/computed tomography are expensive and may not be cost-effective for routine screening [26]. Furthermore, these techniques are associated with a certain frequency of false positives [26]. These factors apply to patients with ADC or SqCC, which we believe are widely applicable regardless of histology.

The time difference in the onset of IR and ER may be explained by the concepts of tumor dormancy [27] and tumor immunoediting theory [28]. According to these concepts, circulating tumor cells are present at the site of recurrence even in the very early stages of tumor progression, but the cells are dormant. Several cell-intrinsic and -extrinsic mechanisms support tumor dormancy during the equilibrium phase of tumor immunoediting [29]. The tumor microenvironment is supported by factors that suppress the immune response, such as regulatory T cells and myeloid-derived suppressor cells (MDSCs), which can break the dormancy of the tumor and cause tumor recurrence [30]. Furthermore, patients with completely resected lung cancer are more prone to recurrence when high numbers of MDSCs are present within the circulation at the time of surgery [31]. If the tumor microenvironment at the site of recurrence is established at the time of surgery, the tumor will immediately recur at the site, whether intra- or extrathoracic. In contrast, tumor cells will remain dormant if the tumor microenvironment is established before surgery. On the basis of our finding that ER was less likely to occur late after surgery, we suggest that the mechanism of dormant tumor cells escaping from the tumor microenvironment in the extrathoracic area is more complex than in the intrathoracic area, making it more difficult to terminate dormancy.

We show here that the characteristics of the hazard curves were similar in all types of histology. We therefore, do not believe that the follow-up program must be drastically adjusted for each histological type. In contrast, it may be necessary to change the follow-up program depending on the tumor pathology. Thus, in stage IA disease, the curve remained low and was essentially flat until 60 months, indicating that frequent examinations may be unnecessary. In stages IB, IIA, and IIB, the curve peaked from 6 to 12 months and decreased until 24 months, followed by a tapered plateau-like tail extending to 60 months, suggesting that frequent examinations may be necessary for the first 24 months after surgery and less frequently thereafter. In stage IIIA, although the curve tended to decrease after peaking at 9 months, it was always higher than those of the other stages, indicating that frequent examinations should be performed for 60 months after surgery. In contrast, when this analysis was limited to patients with SqCC, the recurrence curves for stage IB, stage II, and stage IIIA were similar, suggesting that changes in follow-up methods are not required.

Compared with patients who were not administered adjuvant chemotherapy, the peak recurrence was delayed by approximately 2 to 3 months for patients who received adjuvant chemotherapy, while the peak was not as high except for patients with stage IIIA disease. The shapes of the curves suggest that it is unnecessary to change the follow-up method with or without adjuvant chemotherapy. Thus, we propose that the essential effect of adjuvant chemotherapy may be to delay the peak of recurrence during the first postoperative year by approximately 3 months and to reduce the peak of recurrence of patients with stage IIIA disease.

Together, our results support the conclusion that it is reasonable to propose a follow-up program for efficient and early detection of tumor recurrence in patients after complete resection of lung cancer. The program is stratified using tumor histology and pathological stage (Table 2). Surveillance for IR should be frequently performed for patients with stage IB or higher, and occasionally for patients with stage IA. Furthermore, follow-up examinations of patients with stage IB or stage II may be occasionally performed 24 months after surgery. For patients with stage IIIA ADC, follow-up examinations should be frequently performed, even 24 months after surgery, and occasionally for patients with stage IIIA SqCC 24 months after surgery. In contrast, surveillance for ER should be occasionally performed up to 24 months after surgery and optionally thereafter, regardless of histology or pathological stage of the tumor.

This study has several potential limitations. First, our analysis may have been confounded by the absence of a predetermined, systematic follow-up method, including diagnostic imaging, to evaluate the patterns of recurrence. Thus, the follow-up methods were not standardized because patients’ data were acquired from a database registry comprising the records of numerous medical facilities. Furthermore, the date on which recurrence was confirmed was registered as the date of recurrence. However, recurrence may have occurred between the date immediately before the same test did not detect recurrence and the date on which recurrence was confirmed. We therefore, suspect that deviations from the true recurrence date varied among patients because of inconsistent follow-up systems. Second, to our knowledge, there is no statistical method to compare these two curves. Thus, it was impossible to determine the significance of the differences between them, and we are therefore forced to argue that our conclusions are confirmed by the shapes of the curves. Third, most of the cases in the study were ADC or SqCC, and not enough cases were included to suggest surveillance programs for other histological types. Further studies of minor histology, including small cell carcinoma cases, remain necessary. Moreover, here we considered the optimal follow-up program according to the premise that early detection of lung cancer recurrence prolongs life expectancy, although there is no clear evidence to support this. Considering that the efficacy of chemotherapy recently increased as a result of the post-2010 widespread use of epidermal growth factor receptor tyrosine kinase inhibitors [32,33,34,35,36], anaplastic lymphoma kinase inhibitors [37,38], and immune checkpoint inhibitors [39,40], we assumed that early detection of relapse will prolong life expectancy.

## 5. Conclusions

We found that the risk of recurrence in 12,897 patients who underwent complete resection of lung cancer showed sharp transitions during the long follow-up period. Furthermore, the shapes of the curves and the heights of the peaks varied according to different factors. This hazard function will likely help provide important information that will contribute to the development of individualized, postoperative follow-up programs for patients diagnosed with different disease stages.

## Figures and Tables

**Figure 1 cancers-14-05119-f001:**
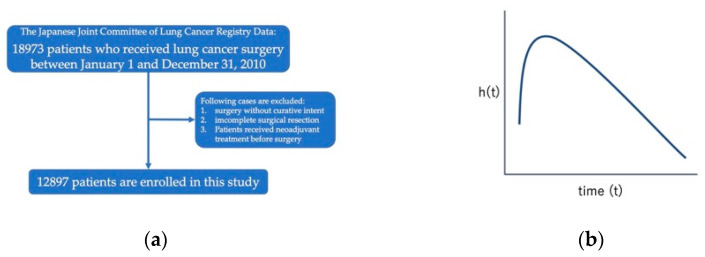
(**a**) Study flowchart. (**b**) Model of hazard function curve.

**Figure 2 cancers-14-05119-f002:**
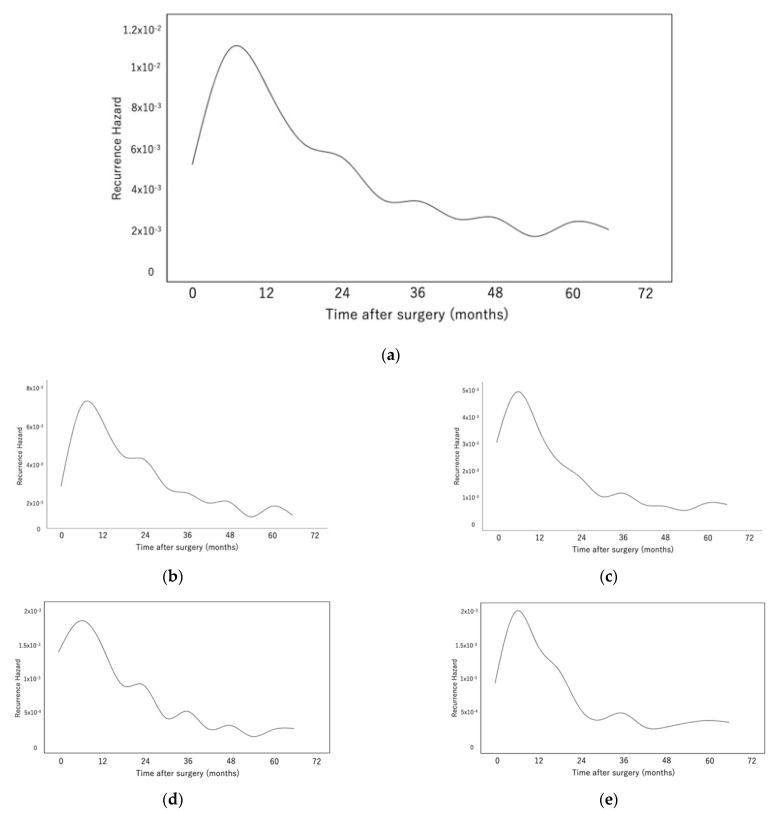
(**a**) The monthly recurrence hazard function of the entire patient population. The major peak of recurrence was 9 months after surgery. (**b**–**e**) The recurrence hazard functions for intrathoracic recurrence (**b**), extrathoracic recurrence (**c**), bone metastasis (**d**), and brain metastasis (**e**).

**Figure 3 cancers-14-05119-f003:**
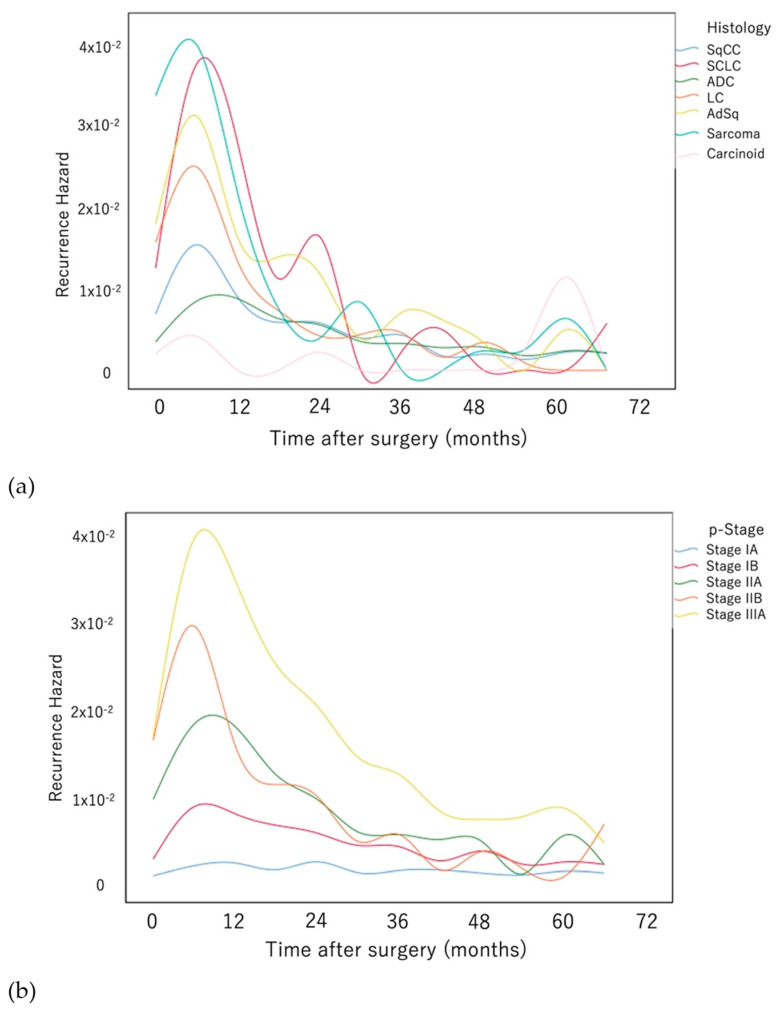
Comparison of the recurrence hazard functions associated with (**a**) histology and (**b**) pathological stage. (Abbreviations: SqCC, squamous cell lung carcinoma; SCLC, small cell lung cancer; ADC, adenocarcinoma; LC, large cell lung carcinoma; AdSq, adenosquamous lung carcinoma.).

**Figure 4 cancers-14-05119-f004:**
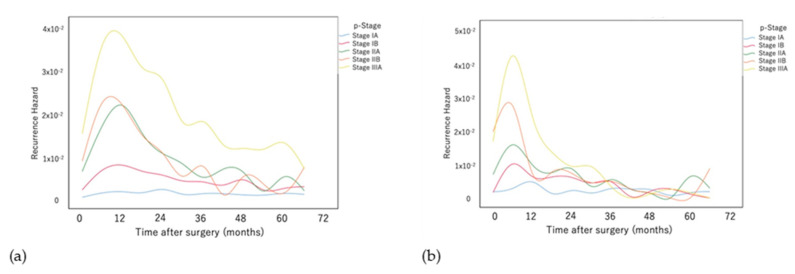
Comparison of the recurrence hazard functions of patients with (**a**) adenocarcinoma and (**b**) squamous cell carcinoma at each pathological stage.

**Figure 5 cancers-14-05119-f005:**
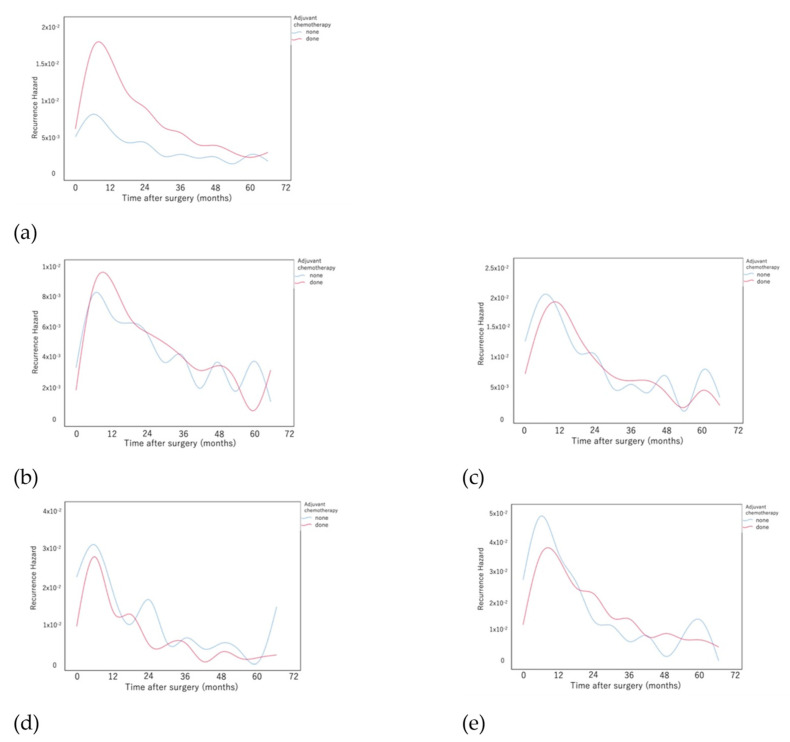
Comparison of the recurrence hazard functions between patients treated with or without adjuvant chemotherapy. (**a**) All patients and patients with (**b**) stage IB, (**c**) stage IIA, (**d**) stage IIB, and (**e**) stage IIIA.

**Table 1 cancers-14-05119-t001:** Patients’ characteristics.

Factors			*N*	(%)
Age	<75		9290	72.0
	≥75		3607	28.0
Sex	Male		8017	62.2
	Female		4880	37.8
ECOG-PS	0		10,758	83.4
	1		1548	12.0
	>2		265	2.1
Smoking history	non-smoker		4490	34.8
	ex-smoker		6093	47.2
	current smoker		1883	14.6
%VC		(%)	107 ± 18	
FEV1/FVC		(%)	74 ± 10	
type of surgery	Pneumonectomy		227	1.8
	bi-lobectomy		335	2.6
	Lobectomy		12,335	95.6
operation time		(min)	222 ± 81	
pathological T factor	T1a		3683	28.6
	T1b		2709	21.0
	T2a		4566	35.4
	T2b		733	5.7
	T3		1066	8.3
	T4		116	0.9
Tumor size		(cm)	3.0 ± 1.7	
pathological N factor	0		10,152	78.7
	1		1307	10.1
	>2		1350	10.5
pathological stage	IA		5585	43.3
	IB		3303	25.6
	IIA		1476	11.4
	IIB		775	6.0
	IIIA		1593	12.4
	IIIB		43	0.3
Histology	Adenocarcinoma		9004	69.8
	Squamous cell carcinoma		2691	20.9
	Large cell carcinoma		398	3.1
	Adenosquamous carcinoma		270	2.1
	Sarcoma		191	1.5
	Small cell carcinoma		179	1.4
	Carcinoid		89	0.7
	Others		75	0.6
pleural infiltration	Negative		9276	71.9
	Positive		3528	27.4
adjuvant therapy	Chemotherapy		4267	33.1
	Radiotherapy		187	1.4

**Table 2 cancers-14-05119-t002:** Proposed optimal follow-up program after complete resection of lung cancer.

Histology	Pathological Stage	Search for Intrathoracic Recurrence		Search for Extrathoracic Recurrence	
		until 24 months	after 24 months	until 24 months	after 24 months
ADC	IA	occasional		occasional	optional
	IB/II	frequent	occasional		
	IIIA	frequent			
SqCC	IA	occasional			
	IB/II	frequent	occasional		
	IIIA	frequent	occasional		

## Data Availability

Data was obtained from the Japanese Joint Committee of Lung Cancer Registry and are available from the authors with the permission of the Japanese Joint Committee of Lung Cancer Registry.

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
