# Peer review of "Hazard Function Analysis of Recurrence in Patients with Curatively Resected Lung Cancer: Results from the Japanese Lung Cancer Registry in 2010"

_cancers, 2022, doi:10.3390/cancers14205119_

Round 1

Reviewer 1 Report

This study tries to investigate the survival analysis through hazard function analysis according to different lung cancer stages based on national cancer register database.

However, this study demonstrated that the possibility of trend time of intra-thoracic recurrence and extrathoracic recurrence between 24 months or after 24 months. This study provides a guidance for physician according to the recurrence risk prediction.

1.      Introduction

Please try to address the lung cancer rate and trend in current era. Due to increasing early lung cancer detection through low-dose CT scan, therefore, lung cancer relapse monitor is more important in this stage.

Related references

2.      Method

Please briefly summarize the study flowchart according to the study design.

Please briefly explain the hazard function with figure. Therefore, the authors could understand more.

3.      Result/Discussion

According to the author’s comment, the recurrence time line changes according to the different AJCC stages, lung cancers. However, the probability of recurrence rate is not well investigated in this study. Please tries to address this issue or listed in the limitation.

If any prediction model including clinical factors and time-line change to predict the cancer relapse.

4.Grammar and spelling

A few minor typos, grammar hiccups should also be corrected, e.g., “fluctuation” in line 134, page 3.

As suggested by the reviewer, we have gone through the revised article and corrected minor typing mistakes and grammatical errors.

Page 3 -Corrected mistake in the “fluctuation” to “fluctuation,” (line 134)

Page 3 -Corrected grammatical mistake in the “curves” to “curve” (line 134)

Page 3 -Corrected mistake in the “squamous cell” to “squamous-cell” (line 142)

Page 5 -Corrected mistake in the “squamous cell” to “squamous-cell” (line 168)

Page 5 -Corrected grammatical mistake in the “a comparisons” to “a comparison” (line 170)

Page 7 -Corrected grammatical mistake in the “regarding” to “about” (line 216)

Page 7 -Corrected grammatical mistake in the “of” to “on” (line 222)

Page 7 -Corrected grammatical mistake in the “treatment protocols” to “treatments protocol” (line 224)

Page 7 -Corrected mistake in the “therefore” to “therefore,” (line 235)

Page 8 -Corrected mistake in the “progression” to “progression,” (line 243)

Page 8 -Corrected grammatical mistake in the “in” to “within” (line 249)

Page 8 -Corrected grammatical mistake in the “for” to “in” (line 257)

Page 8 -Corrected grammatical mistake in the “the those” to “those” (line 266)

Page 8 -Corrected grammatical mistake in the “were” to “was” (line 269)

Page 9 -Corrected grammatical mistake in the “have occurred” to “occurred” (line 297)

Page 9 -Corrected grammatical mistake in the “not possible” to “impossible” (line 301)

Author Response

C1. Please try to address the lung cancer rate and trend in current era. Due to increasing early lung cancer detection through low-dose CT scan, therefore, lung cancer relapse monitor is more important in this stage.

R1. Thank you very much for your valuable comments. We have added the following description to address this:

Lung cancer screening will likely increase the incidence of early-stage lung cancer detected by low‐dose computed tomography (CT), because the effective mass screening of high-risk groups by CT must be beneficial according to a randomized trial[3]. Therefore, an increase in resectable lung cancer cases is anticipated.

C2. Please briefly summarize the study flowchart according to the study design.

R2. We have added a study flowchart as Figure 1a.

C3. Please briefly explain the hazard function with figure. Therefore, the authors could understand more.

R3. We have added an explanation of the hazard function as well as Figure 1b to the revised manuscript in response to your comment.

C4. According to the author’s comment, the recurrence timeline changes according to the different AJCC stages, lung cancers. However, the probability of recurrence rate is not well investigated in this study. Please tries to address this issue or listed in the limitation.

If any prediction model including clinical factors and timeline change to predict the cancer relapse.

R4. The probability of recurrence is shown in the y-axis of these figures. The size of the text in the scale seemed small, and hence we increased it. Moreover, the probability of recurrence in each type could be compared; we described the comparison of this probability in the Discussion (L253-263).

On the other hand, the prediction model including clinical factors is rather different, because there is no suitable statistical analysis between the curves. This point has been already discussed and we have added this as a limitation of the study.

C5. A few minor typos, grammar hiccups should also be corrected, e.g., “fluctuation” in line 134, page 3.

R5. All typographical errors have been corrected in the revised manuscript.

Reviewer 2 Report

The manuscript entitled “Hazard function analysis of recurrence in patients with curatively resected lung cancer: Results from the Japanese Lung Cancer Registry in 2010” describes the recurrence patterns and hazards of the patients with all types of lung cancer who underwent curative surgery.

This retrospective observational study is well organized and important as well as informative, providing the susceptible period for each recurrence pattern.

However, there are some concerns to be solved.

Major:

The study population includes all types of lung cancer who underwent curative surgery.

However, small cell carcinoma (SCLC) is very aggressive and different from the other types in terms of adjuvant chemotherapy regimens, clinical behavior including the recurrence patterns, recurrence sites, and the post-operative monitoring intervals.

And the patients with SCLC are rarely treated with surgery, except for the surgical biopsy of small nodule.

Therefore, I think the monthly recurrence hazard function (Fig. 1), which I think is very important, should be reevaluated in the non-small cell lung cancer (NSCLC) subgroup.

Although the authors have provided the recurrence hazard functions in association with histology in Fig. 2, this result does not mean that the recurrence hazard in the patients with SCLC behave the same as NSCLC.

Minor:

Table 1 needs some improvement by adding ruled lines.

Author Response

Major:

C1. The study population includes all types of lung cancer who underwent curative surgery. However, small cell carcinoma (SCLC) is very aggressive and different from the other types in terms of adjuvant chemotherapy regimens, clinical behavior including the recurrence patterns, recurrence sites, and the post-operative monitoring intervals. And the patients with SCLC are rarely treated with surgery, except for the surgical biopsy of small nodule. Therefore, I think the monthly recurrence hazard function (Fig. 1), which I think is very important, should be reevaluated in the non-small cell lung cancer (NSCLC) subgroup. Although the authors have provided the recurrence hazard functions in association with histology in Fig. 2, this result does not mean that the recurrence hazard in the patients with SCLC behave the same as NSCLC.

R1. Thank you very much for your valuable comments. Only 1.5% of enrolled cases were SCLC; therefore, it is difficult for us to analyze the hazard function in detail, for example, regarding recurrence site or pathological stages, because the number of cases was too small. This situation is also the same in cases with histology other than ADC and SqCC. It seems that Figure 1 represents the characteristics of ADC and SqCC cases because 90% of the study population consisted of these cases. Therefore, we changed the subject of the proposed optimal follow-up program for ADC and SqCC only. Furthermore, we mentioned this point as one of the limitations of the study in the Discussion, as follows: most of the cases in the study were ADC or SqCC, and not enough cases were included to suggest surveillance programs for other histological types. Further studies of minor histology, including small cell carcinoma cases, remain necessary.

Minor:

C2. Table 1 needs some improvement by adding ruled lines.

R2. We have added ruled lines to Table 1 in the revised manuscript.

Round 2

Reviewer 1 Report

This study has been much improved after revision. I have only one comment about the clinical utilization in real-world. Please try to address this hazard function analysis could help early-stage lung cancer patients after LDCT screening program with clinical surveillance plan because still potential relapse odds in early lung cancer post operation in Asian population. Therefore, this study could provide solution in this critical point.

References:

Assessment of selection criteria for low-dose lung screening CT among Asian ethnic groups in Taiwan: from mass screening to specific risk-based screening for non-smoker lung cancer FZ Wu, YL Huang, CC Wu, EK Tang, CS Chen, GY Mar, Y Yen, MT Wu Clinical lung cancer 17 (5), e45-e56

Transl Lung Cancer Res. 2014 Aug; 3(4): 242–249. doi: 10.3978/j.issn.2218-6751.2013.12.05 Recurrence after surgery in patients with NSCLC

Author Response

C1. This study has been much improved after revision. I have only one comment about the clinical utilization in real-world. Please try to address this hazard function analysis could help early-stage lung cancer patients after LDCT screening program with clinical surveillance plan because still potential relapse odds in early lung cancer post operation in Asian population. Therefore, this study could provide solution in this critical point. 

R1. Thank you very much for your valuable comments. We added this point in the discussion part with your raised references.

Reviewer 2 Report

The manuscript has been well revised, and I do not have any other comments.

Author Response

Thank you very much for your review.